# Association between maternal high-risk fertility behaviour and perinatal mortality in Bangladesh: Evidence from the Demographic and Health Survey

Md. Nuruzzaman Khan [1,2]*, Melissa L. Harris[2]

**1** Deartment of Population Science, Jatiya Kabi Nazrul Islam University, Mymensingh, Bangladesh, **2** Centre for Women's Health Research, University of Newcastle, Callaghan, Australia

* sumonrupop@gmail.com, mdnuruzzaman.khan@uon.edu.au

## Abstract

### Background

High-risk fertility behaviours including pregnancy early or late in the reproductive life course, higher parity and short birth intervals are ongoing concerns in Low- and Middle-Income Countries (LMICs) such as Bangladesh. Although such factors have been identified as major risk factors for perinatal mortality, there has been a lack of progress in the area despite the implementation of the Millennium and Sust ainable Development Goals. We therefore explored the effects of high-risk maternal fertility behaviour on the occurrence of perinatal mortality in Bangladesh.

### Methods

A total of 8,930 singleton pregnancies of seven or more months gestation were extracted from 2017/18 Bangladesh Demographic and Health Survey for analysis. Perinatal mortality was the outcome variable (yes, no) and the primary exposure variable was high-risk fertility behaviour in the previous five years (yes, no). The association between the exposure and outcome variable was determined using a mixed-effect multilevel logistic regression model, adjusted for covariates.

### Results

Forty-six percent of the total births that occurred in the five years preceding the survey were high-risk. After adjusting for potential confounders, a 1.87 times (aOR, 1.87, 95% CI, 1.61–2.14) higher odds of perinatal mortality was found among women with any high-risk fertility behaviour as compared to women having no high-risk fertility behaviours. The odds of peri-natal mortality were also found to increase in line with an increasing number of high-risk behaviour. A 1.77 times (95% CI, 1.50–2.05) increase in odds of perinatal mortality was found among women with single high-risk fertility behaviour and a 2.30 times (95% CI, 1.96–2.64) increase in odds was found among women with multiple high-risk fertility behaviours compared to women with no high-risk fertility behaviour.

**Data Availability Statement:** The datasets used and analyzed in this study are available from the Measure DHS website: https://dhsprogram.com/data/available-datasets.cfm.

**Funding:** The author(s) received no specific funding for this work.

**Competing interests:** The authors have declared that no competing interests exist.

## Conclusion

Women's high-risk fertility behaviour is an important predictor of perinatal mortality in Bangladesh. Increased contraceptive use to allow appropriate birth spacing, educational interventions around the potential risks associated with high risk fertility behaviour (including short birth interval) in future pregnacies, and improved continuity of maternal healthcare service use among this population are required to improve birth outcomes in Bangladesh.

## Introduction

An estimated 4.5 million perinatal deaths (stillbirths and early neonatal deaths) occur worldwide every year [1, 2]. Perinatal mortality has been found to be particularly high in low- and middle-income countries (LMICs) and accounts for a large proportion of total child deaths. Among LMICs, the majority of deaths occurs in Sub-Saharan Africa (43 per 1000 live births) and South Asia (41 per 1000 live births) [3, 4]. Particularly in Bangladesh perinatal mortaility accounts for 60% of the total under-five mortality [4, 5]. Despite this, almost all perinatal mortaility are preventable [6, 7]. Avoidable child mortality and its interventions (e.g., improving skilled delivery and neonatal care) are reflected in the Sustainable Development Goals (SDGs), which calls for ending preventable deaths of newborn babies (12 deaths per 1000 live births for neonatal mortality) and children younger than 5 years (25 deaths for 1000 live births for under-five mortality) by 2030 [3]. However, it is unlikely that LMICs such as Bangladesh will achieve these targets unless aggressive country-specific interventions are implemented [6].

Over the past 20 years, considerable improvements have occurred in developing healthcare facilities and accessing maternal healthcare services in LMICs such as Bangladesh, particularly as a result of the Millennium Development Goals period between 2000 to 2015 [8, 9]. As such, the under-five mortality rate has declined significantly. However, this has not occurred for perinatal mortality [1, 2]. For instance, the reduction of perinatal mortality (64 to 41 per 1000 live births) in Bangladesh was much slower than the reduction of under-five mortality (83 to 41 per 1000 live births) between 2000 and 2014 [4, 10]. Women in Bangladesh are mostly not aware of the obstetric complications that prevent timely access to healthcare services and present to hospital with serious obstetric complications when successful intervention is reduced [11]. Availability of critical care for newborns is also limited in Bangladeshi healthcare facilities [12] and where services are available, distance represents a key barrier to access [13]. Consequently, deaths among new borns occur at substantially higher rates in Bangladesh compared to older aged children.

High-risk fertility behaviours, defined as pregnancy occurring early (i.e. <18 years) or later (>34 years) in the reproductive life course, pregnancies of a shorter interval (<24 months) and higher parity (i.e., more than two children) are common in LMICs [14]. The prevalence of early pregnancy (43%), short pregnancy interval (26%), higher parity (24%) are very high in Bangladesh and a significant proportion of pregnancies occur among women with more than one of these specific high-risk behaviours [15–17]. These independent higher-risk fertility behaviour components have been found to interact with each other and adversely affect several maternal health outcomes, including chronic undernutrition and pregnancy complications, which is a risk factor for increased obstetric and newborn complications [14, 18–21]. This along with healthcare facility level challenges may increase the occurrence of perinatal mortaility [14, 18–21].

While it is posited that an association between higher-risk fertility behaviour and perinatal mortality in LMICs such as Bangladesh exists, so far this understanding is limited. Available

studies have mainly reported perinatal mortality outcomes in relation to disadvantaged socio-demographic characteristics (illiteracy, lower-income, and rurality) [4, 22–24]. Only a few available studies in LMICs have determined the effects of particular high-risk fertility behaviours on perinatal mortality, such as, pregnancy before age 18 and short pregnancy interval [4, 22]. A comprehensive understanding of the dynamic nature of perinatal mortality among women with high-risk fertility behaviour in general, and the compounding impact of multiple high-risk fertility behaviours, in particular, are still lacking in LMICs and to date no relevant study has been conducted in Bangaldesh. We, therefore, explored the effect of the high-risk fertility behaviour of mothers on perinatal mortality in Bangladesh adjusting for possible confounders.

## Methods

### Study design and sample

We analysed 2017/18 Bangladesh Demographic and Health Survey (BDHS) data. The BDHS is a nationally representative survey conducted every three years as part of the Demographic and Health Survey Program. The National Institute of Population Research and Training conducted this survey as a local body and representative of the Ministry of Health and Family Welfare of Bangladesh along with a local organization, the Mitra and Associate. Nationally representative households were selected to participate through a two-stage stratified random sampling process. A total of 675 primary sampling units (PSUs) were selected at the first stage of sampling covering each division and urban and rural area separately. The clusters were selected from a list of 293,579 PSUs generated by the Bangladesh Bureau of Statistics as part of the 2011 Bangladesh National Population Census. Three PSUs were excluded due to extreme floods and the survey was conducted in the remaining 672 PSUs. In the second stage of sampling, a fixed number of 30 households were randomly sampled from each of the selected PSUs. Of the 20,160 households selected, the survey was conducted by 19,457 households (96% participation rate). The overall criteria to be included in this survey were: (i) reproductive-aged (15–49 years) married women and (ii) resided permanently in the selected households or were a non-permanent resident of the selected households but resided there on the night before the survey. These criteria were met by 20,376 women. Of these, data were collected from 20,127 women resulting in a response rate of 98.8%. Women were asked to report on their reproductive history, contraception use, conception, pregnancy termination, stillbirths, and live births in the last five years. Women who reported at least one child within five years of the survey date, data on their children aged under five years were also collected. Informed written consent was obtained from all participants. Details of the sampling procedure have been published elsewhere [25].

### Analytical sample

To examine perinatal mortality, this study extracted data from 8,959 pregnancies of 7 or more months gestation (>28 weeks of gestation). Of these, 29 twin births were excluded because of their different mortality risks for both children and mothers [25]. The final analysis sample consisted of 8,930 singleton births.

### Outcome variable

The outcome variable for this study was perinatal mortality (yes, no) defined as the sum of stillbirths (fetal deaths in pregnancies with 7+ [>28 weeks] months of gestation) and early neonatal mortality (deaths within the first week of life). The relevant rate is calculated according to

WHO guidelines.

Perinatal mortality

$$= \frac{Pregnancy\ loss\ with\ 7 + months\ of\ gestation\ (stillbirths) + Deaths\ within\ 7\ days\ of\ live\ births\ (early\ neonatal\ deaths)}{Pregnancy\ loss\ with\ 7 + months\ of\ gestation\ (stillbirths) + Number\ of\ live\ births}$$

$\times\ 1,000$

### Exposure variables

High-risk fertility behaviour (yes, no) was considered the primary exposure variable. The variable was created based on three high-risk fertility markers (with four indicators) as per the relevant international guidelines: (i) age at pregnancy, (ii) interval between indexed pregnancy and the most earlier pregnancy, and (iii) total number of children ever born. A positive response on high-risk fertility behaviour was considered if the pregnancy occurred before age 18 years or after age 34 years, there was less than a 24 month interval between two subsequent pregnancies or if women had at least two children before the indexed child. We also generated a number of sub-variables based on the number of risky behaviors reported by the women. These were: (i) number of high-risk fertility behaviours (no, single, multiple) and no high-risk fertility behaviour (yes, no).

### Covariates

We considered a range of individual-, household-, and community-level factors as potentially influencing the relationship between high-risk fertility behaviour and perinatal mortality. The covariates were selected in two stages. We first generated a list of all potential covariates by reviewing the relevant literature for LMICs, in particular Bangladesh and its neighbouring countries [4, 14, 18–24]. The availability of the selected potential variables was then checked against the variables available for analysis. These variables, as well as the exposure variables, were checked for multicollinearity and highly correlated variables were deleted. Variables that met these criteria were then categorized as individual-, household-, and community-level factors. Individual-level factors were mothers' educational status (no education, primary, secondary and higher) and mothers' working status (yes, no). Household-level factors were mothers' partners education (no education, primary, secondary and higher), partner's occupation (agricultural worker, blue collar worker [rickshaw driver, brick field worker, domestic servant, non-agricultural worker, carpenter, mason etc.), white collar worker [doctor, lawyer, dentists], pink collar worker [big businessmen, small business/trader] and others [not working, unemployed, retired etc.) and wealth quintile (poorest, poorer, middle, richer and richest). Place of residence (urban, rural) and region of the place of residence (Barishal, Chattogram, Dhaka, Khulna, Mymensingh, Rajshahi, Rangpur, and Sylhet) were included as community-level factors. The wealth quintile variable was created by the BDHS using principal component analysis (PCA), which incorporated variables related to household assets, such as roof type or ownership of a radio/television. The PCA generated a score that was divided into five equal parts, with the highest score representing the richest quintile and the lowest score representing the poorest quintile.

### Statistical analysis

Descriptive statistics were used to characterise the respondents. Mixed-effect multilevel logistic regression models were then used to explore the association between the exposure and outcome variables adjusting for individual-, household-, and community-level factors. Mixed effect multilevel logistic regression model was chosen due to the hierarchical structure of the

BDHS data, in which individuals are nested within a household and households are nested within a cluster. Survey data with such a structure creates multiple levels of dependencies, such as respondents from the same household behave similarly and respondents from the same PSU behave similarly. This creates multiple levels of dependencies in the survey data [26]. Both unadjusted and adjusted models were run. In the unadjusted model, a particular exposure variable was considered with the outcome variable whereas in the adjusted model exposure and outcome variable was considered with all covariates. We employed a progressive model building technique and ran a total of three models. Model 1 consisted solely of individual-level factors, while Model 2 incorporated both individual and household-level factors. Finally, Model 3, the final iteration, included individual, household, and community-level factors. The complex survey design was considered in all analyses. Results are reported as Odds Ratios (OR) with 95% Confidence Intervals (95% CI). Statistical package R (version 4.1) was used for all statistical analyses.

### Ethics approval and consent to participate

The survey analysed was approved by the institutional review board of ICF and the National Research Ethics Committee of the Bangladesh Medical Research Council. Informed consent was obtained from all participants. All necessary patient/participant consent has been obtained and the appropriate institutional forms have been archived. No separate ethical approval was required to conduct this study. We obtained permission to access this survey and conduct this research. All methods were performed in accordance with the relevant guidelines and regulations.

## Results

### Participant characteristics

The distribution of the respondents analysed in this study as well as perinatal mortality across individual-, household-, and community-level characteristics is shown in Table 1. The majority of respondents analysed were aged between 20 and 34 years (85.8%) at the indexed pregnancy and possessed secondary level education (48.3%). Around 52% of the total respondents' husbands were blue collar workers, followed by pink (21%) and agricultural workers (20%). More than two-thirds of the respondents were from rural areas and around one-quarter of respondents were from the Dhaka region.

### Distribution of perinatal mortality across individual-, household-, and community-level characteristics

The average rate of perinatal mortality was 48.0 per 1000 births (Table 1). A higher prevalence of perinatal mortality (132.8 per 1000 births) was found among mothers aged >34 years following 52.8 per 1,000 birth among mothers aged <18 years. Prevalence of perinatal mortality was found higher among mothers with primary education (59 per 1,000 births) and those engaged in formal income-generating activities (54 per 1,000 births).

Around 54 per 1,000 perinatal mortality was reported among women whose partners were blue collar workers. Prevalence of perinatal mortality was found to be higher among mothers in the poorest wealth quintile (52 per 1,000 birth) followed by the poorer wealth quintile (50 per 1,000 births) and the middle wealth quintile (51 per 1,000 births). A similar prevalence of perinatal mortality was reported between mothers from rural and urban areas although Chattogram and Rajshahi regions had the highest prevalence of perinatal mortality with around 53 deaths per 1,000 births.

**Table 1. Sample characteristics of number of pregnancies of 7+ months duration and distribution of perinatal mortality (per 1000 births) in Bangladesh, 2017/18.**

| Characteristics | Prevalence, 95% CI | Perinatal mortality per 1000 births (95% CI) |
|---|---|---|
| **Overall** | | **48.0 (43.0–53.4)** |
| **Mothers age at the indexed birth** | | |
| <18 | 9.4 (8.7–10.2) | 52.4 (38.1–71.7) |
| 20–34 | 85.8 (84.9–86.6) | 42.7 (37.9–48.2) |
| ≥35 | 4.8 (4.3–5.3) | 132.8 (98.5–176.6) |
| **Mother's educational status** | | |
| No education | 7.3 (6.5–8.3) | 40.1 (25.7–62.1) |
| Primary | 29.1 (27.4–30.7) | 58.6 (48.7–70.3) |
| Secondary | 48.3 (46.7–49.9) | 45.3 (38.7–53.0) |
| Higher | 15.3 (14.1–16.6) | 39.9 (29.7–53.4) |
| **Mother's working status** | | |
| Yes | 40.8 (38.7–42.9) | 53.7 (45.6–63.3) |
| No | 59.2 (57.1–61.3) | 44.0 (38.3–50.5) |
| **Partner's occupation** | | |
| Agricultural worker | 20.3 (18.8–21.9) | 43.6 (34.8–54.6) |
| Blue color worker | 52.1 (50.4–53.8) | 53.5 (46.5–61.5) |
| White color worker | 5.6 (4.9–6.3) | 34.8 (19.7–60.9) |
| Pink color worker | 21.1 (19.9–22.4) | 41.4 (32.4–52.7) |
| Others | 0.94 (0.7–1.24) | 50.9 (16.5–146.2) |
| **Wealth quintile** | | |
| Poorest | 21.5 (19.4–23.7) | 51.9 (41.9–64.1) |
| Poorer | 20.4 (19.0–21.8) | 49.6 (39.7–61.7) |
| Middle | 18.8 (17.5–20.2) | 50.3 (39.4–64.1) |
| Richer | 20.1 (18.6–21.8) | 48.9 (38.5–61.9) |
| Richest | 19.2 (17.5–20.9) | 38.5 (29.2–50.7) |
| **Place of residence** | | |
| Urban | 27.4 (25.9–28.9) | 48.6 (39.9–58.9) |
| Rural | 72.6 (71.1–74.1) | 47.7 (41.9–54.4) |
| **Region of the place of residence** | | |
| Barishal | 5.5 (5.1–6.1) | 45.3 (33.3–61.3) |
| Chattogram | 20.9 (19.5–22.5) | 52.8 (41.6–66.7) |
| Dhaka | 25.9 (24.3–27.5) | 47.2 (37.0–60.0) |
| Khulna | 9.1 (8.4–9.9) | 40.5 (26.2–62.0) |
| Mymensingh | 8.4 (7.7–9.2) | 51.9 (40.5–66.3) |
| Rajshahi | 11.7 (10.6–12.9) | 53.3 (39.7–71.2) |
| Rangpur | 10.3 (9.5–11.2) | 44.8 (30.8–64.9) |
| Sylhet | 8.1 (7.3–9.1) | 40.6 (30.8–53.3) |

## High-risk fertility behaviour

Of the women's data analysed, around 46% of the births in the five years preceding the survey were in any-high risk category (Table 2). Of them, a single high-risk fertility behaviour was reported by 38% of total respondents and multiple high-risk fertility behaviour was reported by 8%. The most common form of single high-risk fertility behaviour was birth order above 2 (>2 children ever born, 61%) following pregnancy aged less than 18 years (23%) and a birth interval of less than 24 months (14%). The most common form of multiple high-risk fertility behaviour was pregnancy aged above 34 years and birth order above 2 (48%) followed by birth

**Table 2. Pattern of higher risk fertility behaviour and distribution of perinatal mortality across pattern of higher risk fertility behaviour in Bangladesh, 2017/18.**

| | High risk fertility behaviour | | Perinatal mortality | |
| --- | --- | --- | --- | --- |
| | Weighted frequency | Prevalence, 95% CI | Weighted frequency | Prevalence, 95% CI |
| **High risk fertility behaviour** | | | | |
| No | 4816 | 53.9 (52.5–55.4) | 80 | 1.7 (1.3–2.1) |
| Yes | 4115 | 46.1 (44.7–47.5) | 349 | 8.5 (7.5–9.5) |
| **Single high risk fertility behaviour** | **3378** | **37.8 (36.5–39.2)** | **254** | **7.5 (6.6–8.6)** |
| Age less than 18 years | 765 | 22.7 (21.03–24.4) | 11 | 1.38 (0.06–3.0) |
| Age above 34 years | 56 | 1.7 (1.3–2.2) | 2 | 3.6 (0.07–14.7) |
| The birth interval of less than 24 months | 479 | 14.2 (12.5–16.0) | 55 | 11.4 (8.5–15.1) |
| Birth order above 2 | 2078 | 60.8 (58.6–63.0) | 165 | 8.0 (6.8–9.5) |
| **Multiple high risk fertility behaviour** | **737** | **8.3 (7.5–9.1)** | **94** | **12.8 (10.0–16.2)** |
| Age less than 18 years and birth interval less than 24 months | 54 | 7.3 (4.9–10.8) | 12 | 21.6 (12.6–34.7) |
| Age above 34 years and birth interval less than 24 months | 4 | 0.1 (0.01–1.89) | 1 | 4.4 (0.04–32.0) |
| Age above 34 years and birth order above 2 | 354 | 48.0 (43.4–52.7) | 55 | 15.4 (11.4–20.6) |
| Age above 34 years and birth interval less than 24 months, and birth order above 2 | 14 | 2.0 (0.09–4.2) | 0 | 0 |
| Birth interval less than 24 months and birth order above 2 | 311 | 42.2 (37.5–47.0) | 28 | 8.9 (5.4–14.3) |
| **No high-risk fertility behaviour** | 2402 | 26.9 (25.9–27.9) | 53 | 2.2 (1.6–3.0) |

interval less than 24 months and birth order above 2 (42%). Prevalence of perinatal mortality was found to be higher among women with any high risk fertility behaviour (8.5% *vs* 1.7%). The prevalence was found to be even higher among women with multiple high risk fertility behaviours (12.8%) than single high risk fertility behaviour (7.5%).

## Relationship between high-risk fertility and perinatal mortality

The results of the mixed-effect multilevel logistic regression model to explore the association of perinatal mortality with high-risk fertility behaviour, the number of high-risk fertility behaviour, and no high-risk fertility behaviour are presented in Table 3 and S1–S3 Tables. The

**Table 3. Unadjusted and adjusted associations of perinatal mortality with high-risk fertility behaviour, number of high-risk fertility behaviour, unavoidable high-risk fertility behaviour, and no high-risk fertility behaviour, Bangladesh 2017/18.**

| Characteristics | Unadjusted association | | Adjusted association[++] | |
| --- | --- | --- | --- | --- |
| | OR (95% CI) | *p-value* | OR (95% CI) | *p-value* |
| **High risk fertility behaviour** | | | | |
| No (ref) | 1.00 | | 1.00 | |
| Yes | 5.83 (4.52–7.52) | 0.000 | 1.87 (1.61–2.14) | 0.000 |
| **Number of high-risk fertility behaviour** | | | | |
| No (ref) | 1.00 | | 1.00 | |
| Single | 5.37 (4.13–6.97) | 0.000 | 1.77 (1.50–2.05) | 0.000 |
| Multiple | 8.00 (5.78–11.09) | 0.000 | 2.30 (1.96–2.64) | 0.000 |
| **No high-risk fertility behavior** | | | | |
| No (ref) | 1.00 | | 1.00 | |
| Yes | 0.39 (0.29–0.52) | 0.000 | 0.36 (0.27–0.49) | 0.000 |

**Notes:** Models are adjusted with respondents' education, respondents' works, wealth quintile, respondents' husband occupation, place of residence and region of place of residence

likelihood of perinatal mortality was found to be 1.87 times higher (aOR, 1.87, 95% CI, 1.61–2.14) among mothers with any high-risk fertility behaviours as compared to the mothers without any high-risk fertility behaviour. Compared to the women with no high risk fertility behavior, the likelihood of perinatal mortality was found to be higher among women with single high-risk fertility behaviour (aOR, 1.77, 95% CI, 1.50–2.05) and multiple high-risk fertility behaviours (aOR, 2.30, 95% CI, 1.96–2.64). Additionally, likelihood of perinatal mortality was found 64% (aOR, 0.27–0.49) lower among mothers having no high-risk fertility behaviour as compared to the mothers having any high-risk fertility behaviour.

## Discussion

In our analysis of 2017/18 BDHS data, we found that around 46% of the total births that occurred within five years preceding the survey were in any-high risk category, including 38% and 8% of the total births which were in the single and multiple high-risk categories, respectively. The results demonstrate that women's high-risk fertility behaviour increases the occurrence of perinatal mortality in general and is further compounded with increasing numbers of high-risk fertility behaviours. This indicates that attaining the SDG targets related to neonatal and under-five mortality are improbable for Bangladesh where around half of the total births occur among women whose pregnancies involve single or multiple high-risk fertility behaviours. These findings highlight the need for national-level policies and programs to reduce pregnancy with high-risk fertility behaviours of mothers and continuity of maternal healthcare services use.

The prevalence of high-risk fertility behaviour found in this study for Bangladesh is higher than the other south Asian countries [18, 27], but lower than in African countries [28]. However, an important observation is that the prevalence of high-risk fertility behaviour reported in this study is higher than the previously reported prevalence for Bangladesh (36%) [10]. A similar rising pattern has been observed in other LMICs. This is concerning considering significant public health measures have been taken in LMICs to improve maternal and child health, particularly the MDGs period which ended only three years prior to the time of this survey [29]. This rising prevalence could be linked with the recent patterning of fertility in LMICs such as Bangladesh. For instance, total fertility rate was high in Bangladesha decade ago, as such there remains a higher percentage of reproductive-aged women [30]. Among these women, early marriage and early child-bearing are still highly prevalent and due to such early initation into reproductive life, these women have a longer period of child-bearing [31]. This presents issues such as higher parity and close birth spacing. Despite this, pregnancy at later ages is also increasing rapidly in LMICs such as Bangladesh due to delayed starting of conjugal life, mostly because of the increasing age at marriage, women's empowerment, and higher prevalence of unmet need for family planning and contracrption [32]. These issues are directly contributing to the rising prevalence of high-risk fertility in Bangladesh.

In our analysis, the odds of perinatal mortality was found to be around 2 times higher among women with any high-risk fertility behaviours after controlling for potential confounders and this was found to increase further with the number of high-risk fertility behaviours. These observations are particularly novel for LMICs. As such, we are unable not make direct comparisons. However, available studies in LMICs and Bangladesh have found individual high-risk fertility to be linked with several adverse child health outcomes, including under-five mortality [14, 21]. Importantly, each type of high-risk fertility behaviour has been linked to numerous adverse characteristics which later lead to the occurrence of perinatal mortality.

For instance, women who become pregnant at an early age may not have matured physically to allow for optimal foetal development [21, 33] and they may not aware of their

upcoming child health issues [34]. This also increases the occurrence of unintentional injury [35]. Moreover, low maternal healthcare service use and in particular continuity of care has been found to be higher among younger and older aged mothers [36]. This has been attributed to factors such as inadequate knowledge about pregnancy, the importance of maternal healthcare services use, lower empowerment, lower autonomy to access maternal healthcare services and tendency to depend on prior pregnancy experiences [36, 37]. Because of these charecteristics, as well as lower and higher age of women, are also found to be associated with experiencing unintended pregnancy, which accounts for a quarter of live births in Bangladesh [31]. Unintended pregnancy also contributes numerous adverse consequences and lower use of maternal healthcare services [31]. Also, pregnancy among women later in their reproductive lives are at increased risk for congenital and chromosomal abnormalities [21, 33]. Short interval birth could also be linked with perinatal mortality in several ways. For instance, mothers with a short birth interval cannot recover their nutritional stores which may result in malnutrition in the next pregnancy [38]. It also increases the risks of placenta abruption and premature rupture of the membrane [39]. Higher parity, among which the occurrence of birth in short interval, is associated with maternal undernutrition and lower access or maternal healthcare services [40]. These characteristics thereby increase the risk of adverse pregnancy outcomes, including anaemia, intrauterine growth retardation, preterm birth, and low birth weight [14, 21, 34, 41]. Importantly, the extent of these adverse outcomes increase with the increasing number of high-risk ferility behaviours and for particular risk factors such as early or late aged pregnancy. Such lower use of healthcare services use along with continuity, higher occurrence of unwanted pregnancy as well as other adverse charecteristics later contribute to increase occurence of perinatal mortality.

To address the burden of high-risk fertility behaviour and related adverse consequences in Bangladesh, prevention, protection and treatment strategies need to be focused in the relevant policies and programs which are currently absent. For this to happen, mothers in any of the high-risk categories should be counselled appropriately about their potential risk of getting pregnancy and having baby if they became pregnant and have a baby. Improving the uptake of maternal and postnatal healthcare services, along with disaparity in availability and accessing of maternal healthcare services by urban vs rural and place of residence, are also key to the prevention of neonatal and infant mortality. Access to family planning services and use of contraception, including access to contraception at the post-partum period, should also be ensured, particularly among the disadvantaged and unaware population, as well as the rural population, where occurrence of birth in short interval is very higher [16, 42]. Existing health problems, including maternal undernutrition and anaemia, are required to be addressed properly. For this, screening programs need to be ensured before pregnancy through community level family planning workers as well as during pregnancy when women accessing preconception care and treat them. Finally, maternal healthcare services, as well as the continuity of using maternal healthcare services, should be ensured for all women, particularly, for women having any high-risk fertility behaviour.

The findings of this study should be interpreted while taking into account its limitations. First, the findings of this study were generated based on analysing a cross-sectional survey, therefore, they are correlational only rather than casual. Second, data recorded in the survey analysed were collected based on the mothers' retrospective responses to the events that occurred up to five years back. This increased the risk of recall bias, although during data collection the respondents were asked multiple questions which helped the survey to reduce the recall bias. Third, this study does not incorporate healthcare facility-level factors in the models, though a broad range of individual, household and community level covariates were adjusted. Since healthcare facilities place an important role in the occurrence of perinatal mortality,

such a lack could deter the strengths of the reported association between exposure and outcome variable. Similarly, mothers' physical and behavioural characteristics at the time of indexed pregnancy and maternal healthcare services use as well as continuity of using maternal healthcare services are important factors to consider in the model to determine a precise estimate of the reported association. Also, this study focused exclusively high-risk behaviours among women. As women's partners play a key role in family planning (or lack thereof), understanding the role partner attitudes regarding fertility and family planning are needed to in future studies. Despite this, the findings of this study are robust as they were generated by analysing nationally representative survey data with a quite large sample size. Advanced statistical modelling technique was used to determine the targeted associations. Therefore, the findings of this study will be helpful for evidence-based policy and program making. This would ultimately help LIMCs and Bangladesh in reducing the prevalence of high-risk fertility behaviour and perinatal mortality which are imperative to achieving the relevant SDGs targets by 2030.

## Conclusions

High-risk fertility behaviour was reported by almost half of mothers in this study with, 38% reporting single high-risk fertility behaviour and 8% reporting multiple high-risk fertility behaviour. We found a high odds of perinatal mortality among mothers having any high-risk fertility behaviours and the odds were found to be increased with the increased number of high-risk fertility behaviour of mothers. Such a higher prevalence of high-risk fertility behaviour along with increased perinatal mortality among this population indicate it is improbable for Bangladesh to achieve SDG targets of reducing neonatal and under-five mortality to 12 and 25 per 1000 live births by 2030. Maternal healthcare services use with continuity are important to reduce occurrence of high-risk ferility and related adverse outcomes including perinatal mortality. Policy and program development is also needed reduce the prevalence of high-risk fertility behaviour through education about the adverse effects of high-risk fertility behaviour as well as ensuring proper family planning services and contraception in preconception and post-partum period. These should be an important pathway to achieving the SDGs goals by 2030 and ensuring positive outcomes for both women and children.

## Supporting information

**S1 Table. Association between perinatal mortality and high risk fertility behaviour adjusted for individual, household and community level factors.**
(DOCX)

**S2 Table. Association between perinatal mortality and number of high-risk fertility behaviour adjusted for individual, household and community level factors.**
(DOCX)

**S3 Table. Association between perinatal mortality and no high risk fertility behaviour adjusted for individual, household and community level factors.**
(DOCX)

## Acknowledgments

We acknowledge the DHS of the USA for approving us to use their data. Melissa L Harris is supported by a Glady M. Brawn Fellowship.

## Author Contributions

**Conceptualization:** Md. Nuruzzaman Khan.

**Data curation:** Md. Nuruzzaman Khan.

**Formal analysis:** Md. Nuruzzaman Khan.

**Methodology:** Md. Nuruzzaman Khan.

**Resources:** Md. Nuruzzaman Khan.

**Software:** Md. Nuruzzaman Khan.

**Supervision:** Melissa L. Harris.

**Writing – original draft:** Md. Nuruzzaman Khan.

**Writing – review & editing:** Md. Nuruzzaman Khan, Melissa L. Harris.

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
