## [Decision Letter · Decision Letter 0]

4 Oct 2023

PONE-D-23-23194Association between maternal high-risk fertility behaviour and perinatal mortality in Bangladesh: evidence from the Demographic and Health SurveyPLOS ONE

Dear Dr. Khan,

Thank you for submitting your manuscript to PLOS ONE. After careful consideration, we feel that it has merit but does not fully meet PLOS ONE’s publication criteria as it currently stands. Therefore, we invite you to submit a revised version of the manuscript that addresses the points raised during the review process.

The manuscript has been evaluated by two reviewers, and their comments are available below. The reviewers were positive about the study, but they have raised a few concerns that need attention, particularly with respect to presentation of the analysis and statistics. Could you please revise the manuscript to carefully address the concerns raised?

We look forward to receiving your revised manuscript.

Kind regards,

Marianne Clemence

Staff Editor

PLOS ONE

Reviewers' comments:

Reviewer's Responses to Questions

**Comments to the Author**

1. Is the manuscript technically sound, and do the data support the conclusions?

Reviewer #1: Yes

Reviewer #2: Yes

2. Has the statistical analysis been performed appropriately and rigorously? 

Reviewer #1: Yes

Reviewer #2: Yes

3. Have the authors made all data underlying the findings in their manuscript fully available?

Reviewer #1: No

Reviewer #2: Yes

4. Is the manuscript presented in an intelligible fashion and written in standard English?

Reviewer #1: Yes

Reviewer #2: Yes

5. Review Comments to the Author

Reviewer #1: The article is very well-written, provides background information in a good manner, methods and results are explained well and discussed accordingly. I have several points to highlight:

1. It is always important to know what are the basis of choosing the confounders, especially when it comes for the outcome as PM, I would prefer to have a paragraph in the methods explaining the reasons of choosing one or another confounder for adjusting. I also think age is an important factor to take into account, rather than working status, which is affected by age.

2. The PM rate needs to be counted per 1000 births and not live births, its important and please have a close look at the manuscript for each place authors are referring PM rate.

3. In my mind, there are lack of discussion on characteristics, while there are many things presented in characteristics table in results, it would be better for article quality if authors try and explain some of the differences in PM rates for different characteristics, like region.

Overall, I think the research is well-conducted and needs to be published.

Reviewer #2: This is a manuscript with a relevant theme for public health, with a clear and objective presentation, current theoretical framework. It presents a well-contextualized introduction section and a clear objective. In the method section, it brings the necessary details for understanding the research scenario, data collection and study variables, as well as presenting the data analysis with consistency between the chosen tests.

In the result section, clear texts and with notes without repetitions of everything that is in the tables. The tables are well organized, but I suggest just including the year and place in the description.

As for the discussion section, the authors bring counterpoints in relation to the evidenced results, leaving a series of reflections to the reader about the findings; important reflections so that one can think of strategies for discussions about public policies and decision-making.

In conclusion, it responds to the proposed objective and brings important inferences that reveal the relevance of the study.

6. PLOS authors have the option to publish the peer review history of their article (what does this mean?). If published, this will include your full peer review and any attached files.

Reviewer #1: **Yes: **Tinatin Manjavidze

Reviewer #2: No

---

## [Author Response · Author response to Decision Letter 0]

11 Oct 2023

We have added a MS word file addressing each comment.

---

## [Decision Letter · Decision Letter 1]

2 Nov 2023

Association between maternal high-risk fertility behaviour and perinatal mortality in Bangladesh: evidence from the Demographic and Health Survey

PONE-D-23-23194R1

Dear Dr. Khan,

We’re pleased to inform you that your manuscript has been judged scientifically suitable for publication and will be formally accepted for publication once it meets all outstanding technical requirements.

Kind regards,

Pintu Paul

Academic Editor

PLOS ONE

Additional Editor Comments (optional):

Reviewers' comments:

Reviewer's Responses to Questions

**Comments to the Author**

1. If the authors have adequately addressed your comments raised in a previous round of review and you feel that this manuscript is now acceptable for publication, you may indicate that here to bypass the “Comments to the Author” section, enter your conflict of interest statement in the “Confidential to Editor” section, and submit your "Accept" recommendation.

Reviewer #1: All comments have been addressed

2. Is the manuscript technically sound, and do the data support the conclusions?

Reviewer #1: Yes

3. Has the statistical analysis been performed appropriately and rigorously? 

Reviewer #1: Yes

4. Have the authors made all data underlying the findings in their manuscript fully available?

Reviewer #1: No

5. Is the manuscript presented in an intelligible fashion and written in standard English?

Reviewer #1: Yes

6. Review Comments to the Author

Reviewer #1: The responses on the questions/comments were absolutely understandable and I recommend for publication.

7. PLOS authors have the option to publish the peer review history of their article (what does this mean?). If published, this will include your full peer review and any attached files.

Reviewer #1: No

---

## [Editor Report · Acceptance letter]

13 Nov 2023

PONE-D-23-23194R1 

Association between maternal high-risk fertility behaviour and perinatal mortality in Bangladesh: evidence from the Demographic and Health Survey 

Dear Dr. Khan:

I'm pleased to inform you that your manuscript has been deemed suitable for publication in PLOS ONE. Congratulations! Your manuscript is now with our production department. 

Kind regards, 

on behalf of

Dr. Pintu Paul 

Academic Editor

PLOS ONE